# Traditional Yerba Mate Agroforestry Systems in Araucaria Forest in Southern Brazil Improve the Provisioning of Soil Ecosystem Services

Lucilia M. Parron [1,*], Ricardo Trippia dos G. Peixoto [2] , Krisle da Silva [1] and George G. Brown [1]

[1] Embrapa Forestry, Estrada da Ribeira, km 111, Caixa Postal 319, Colombo 83411-000, PR, Brazil; krisle.silva@embrapa.br (K.d.S.); george.brown@embrapa.br (G.G.B.)
[2] Embrapa Soils, Rua Jardim Botanico, 1024, Rio de Janeiro 22460-000, RJ, Brazil; ricardo.trippia@embrapa.br
[*] Correspondence: lucilia.parron@embrapa.br

**Abstract:** Soils are a source of natural capital that provide and regulate a range of ecosystem services (ES) and play an important role in sustaining human welfare. Nonetheless, the quality and quantity of soil ecosystem services (SES) delivery over the long term depend on the use of sustainable land management practices. In the present study, we assessed seven SES using a set of soil quality indicators in four production systems based on yerba mate (*Ilex paraguariensis* A. St.-Hil) in the Araucaria Forest biome of Southern Brazil: two sites were under traditional agroecological agroforestry management, one was a silvopastoral system with dairy pasture, and the last one was a monoculture yerba mate production system. The SES measured were soil fertility, carbon sequestration, erosion control, nutrient cycling, plant provision, biodiversity, and health. Soil samples were collected at various depths and analysed for chemical, physical, and biological attributes. A principal component analysis on the dataset showed that the soil quality indicators that best represent the variance between the systems at the 0–10 cm layer were acidity, microbial activity (FDA), total nitrogen, (TN), structural stability index (SSI), cation exchange capacity (CEC), pH, sum of bases (SB), microbial quotient (qMic), density of earthworms (EwD), bulk density (BD), and carbon stocks ($C_{stock}$). Soil quality indicators ranging from 0 to 1 were used to graphically represent the set of SES. The indicator-based approach used to explain the differences among the four production systems was able to capture the soil functions and offered a good starting point for quantifying SES provision.

**Keywords:** ecosystem services indicators; principal components analysis; soil attributes; soil conservation; soil quality; traditional and agroecological yerba mate





## 1. Introduction

Ecosystem services (ES) are generally defined as "benefits that people obtain from ecosystems" [1]. These include provisioning services such as food and water; regulating services such as flood and disease control; cultural services such as spiritual, recreational and cultural benefits; and supporting services such as nutrient cycling that maintain the conditions for life on Earth [2]. The ES concept offers a way to understand and perhaps deal with the negative feedback loop that is ultimately created when the ecosystems are used for human needs [3–5].

ES are affected by landscape heterogeneity and, despite increasing knowledge of the impacts of human activities on the environment, landscapes continue to be transformed in unsustainable ways to meet our needs for food, water, fuel, shelter, etc. [6]. Although multifunctional landscapes are often more resilient to ecosystem shocks and disturbances, such as deforestation, or climate-induced environmental variation [7], land use changes impact ecosystem functions and services provided by those landscapes [8], for instance, soil ES (SES). By understanding the effect of human activities on ES, the analysis of land and landscape management decisions contributes to identifying optimal ES provisioning.

Soil is an important environmental component (natural capital), responsible for providing support and regulation for a large number of ES and plays an important role in human welfare [9]. SES can be provided by agricultural soils and there have been many studies defining the linkage of soil properties to ES, including global reviews [10–12] and methodological frameworks [13–17]. Incorporating the contribution of SES in landscape management can be achieved by linking them to the multitude of functions it provides [14,18,19].

SES delivery depends on soil properties and their interaction and are mostly influenced by its use and management. Erosion, for example, reduces soil carbon stocks and biodiversity, leading to soil degradation, which is a serious global challenge for food security and ecosystem sustainability [10]. As most SES include multiple functions, their quantification is not simple. Hence, certain functions and properties that are more readily measurable, and that have strong linkages to the services in question, can be used as indicators or proxies of SES [14,20].

Quantifying ES can be a major challenge, and some researchers have made thorough analyses using mathematical representations and statistical methods to model SES [17,21], while others have used GIS-based models that aim to value multiple ES across a landscape in different land-use scenarios [22]. Overall, to truly realise and model the complexity of ES trade-offs and to be able to apply the ES concept in management decisions, improved quantification methods are needed [23].

The Brazilian Atlantic Forest is a global biodiversity hotspot [24] that provides important ES to a large proportion of the Brazilian population [25]. Within the Atlantic Forest, many useful plants can be found, including fruits, as well as a widely used beverage plant called yerba mate (*Ilex paraguariensis* A. St.-Hil.) [26]. This non-timber forest species is highly appreciated in the Central-Southern region of Brazil, primarily for use in beverages like *chimarrão*, *tererê*, and mate tea. The drink is also widely enjoyed in Argentina, Paraguay, and Uruguay. However, the popularity of yerba mate is expanding into new markets, such as the USA, Europe, and Asia, due to its high antioxidant content and proven health benefits, as well as its use in energy drinks [27]. The distribution of this species predominantly aligns with the subtropical *Araucaria* Forest [*Araucaria angustifolia* (Bertol.) Kuntze], covering an area of around 450,000 km$^2$ [26].

Yerba mate evolved in an integrated way with several forest species, being tolerant to low temperatures and shading, in soils with low pH, high levels of aluminium, and low levels of exchangeable cations, showing symbiosis with arbuscular mycorrhizal fungi [26]. The yerba mate production occurs in monoculture systems under full sun exposure, as well as in agroforestry systems with traditional and agroecological management (mostly without fertilisation) [28,29], and in silvopastoral systems known as *Caíva* [30], with yerba mate growing under the *Araucaria* Forest canopy.

In this context, the production of yerba mate in traditional agroforestry systems in southern Brazil, based on culturally accumulated community knowledge, has contributed to the conservation of Araucaria Forest remnants [29]. When well-managed, they can contribute to sustainable land use, soil protection, water security, biodiversity, and the rural environment, as well as towards SES [31–34]. However, these SES have not yet been valued for economic compensation. Little information is available regarding the relationship between yerba mate production systems and soil quality, aiming to conserve and improve soil multifunctionality, which increase the provision of ES, as well as the sustainability of these agroforestry production systems.

Therefore, a case study was carried out in four yerba mate production systems maintained by family farmers in Paraná State, with the aim of characterising and selecting the most efficient soil attributes to evaluate soil quality, and their relationships with the provisioning of SES.

## 2. Material and Methods

### 2.1. Study Area and Sampling

The experimental areas were located in the Bituruna region, Paraná State, Southern Brazil, in a region of *Araucaria* Forest (mixed evergreen and deciduous mountain forest, or *Floresta Ombrófila Mista Montana* within the Atlantic Forest biome, according to the Brazilian classification [35]. The climate is subtropical, classified as Cfb according to Köppen [36]. The annual rainfall is 1700 mm yr$^{-1}$, with average annual temperatures between 15 °C and 18 °C [37]. The sites lie on Cambisols, consisting of shallow soils with a clayey to very clayey texture [38].

The state of Paraná is the main producer of yerba mate in Brazil. In 2021, Paraná accounted for 87% (443 thousand tons) of Brazilian production (506 thousand tons), and Bituruna was responsible for 11% of the national production. The value of yerba mate production in Paraná was BRL 684 million and Bituruna's contribution amounted to BRL 70 million in 2021. Hence, the study area is one of the main producing regions of yerba mate in Brazil, contributing to both the local and national economy, as well as to farmer and industrial income generation [39].

Four sites with yerba mate production were defined for comparisons (Table 1 and Figure 1). Each site was located less than 2 km from each other and had similar soil conditions, with undulating relief, moderate slopes of 8 to 20%, and different land uses. Two sites were under traditional agroecological agroforestry management, one was a silvopastoral system with dairy pasture, and the last one was a monoculture yerba mate production system. Detailed descriptions of the study sites are given in [40,41].

**Table 1.** Selected details concerning the four yerba mate production systems evaluated in the present study, in the region of Bituruna, Paraná State, Brazil.

| | Acronym | Altitude (m) | Geographic Coordinate | Yerba Mate Density (Trees ha$^{-1}$) * | Native Tree Density (Trees ha$^{-1}$) * | Notes |
|---|---|---|---|---|---|---|
| Traditional Agroecological System in *Araucaria* Forest (Agroforestry System) | AFS-A | 1.030 | S 26°12′6.442″ O 51°26′32.288″ | 2000 | 388 | Area with yerba mate planted densely about 18 years ago, among native yerba mate and secondary forest in medium stage of ecological succession dominated by pioneer and early secondary forest species. No chemical inputs. |
| | AFS-B | 930 | S 26°10′8.249″ O 51°21′55.547″ | - | 940 | Area with yerba mate planted densely about 15 years ago, among native yerba mate aged between 50 and 100 years, under forest vegetation that is older and denser than in AFS-A, in the middle secondary stage with dominance of *Araucaria*, together with pioneer and secondary species. No chemical inputs. |
| Traditional Silvopastoral System in *Araucaria* Forest (*Caíva*) | SPS | 807 | S 26°10′8.249″ O 51°21′55.547″ | - | 236 | Area with 20 to 50 years of integrated management of native yerba mate in forest fragment with less dense *Araucaria*, and dairy cattle in pasture, with liming. |
| Yerba mate in Monoculture System | MCS | 938 | S 26°11′1.197″ O 51°22′13.483″ | 3.133 ** | 200 | Area with yerba mate planted in monoculture with high density, and under full sun, for about 20 years, with management and application of inputs, such as liming and mineral fertilisers. |

* Tree density data (yerba mate and natives) at the sites [41]. Native tree density (trees ha$^{-1}$) = number of native trees per hectare, with sampling error between 21.8% and 77.4%. ** Calculated from the spacing of 2 × 1.5 m, and by subtracting the native tree density (trees ha$^{-1}$) [41].

AFS-A (Agroforestry)

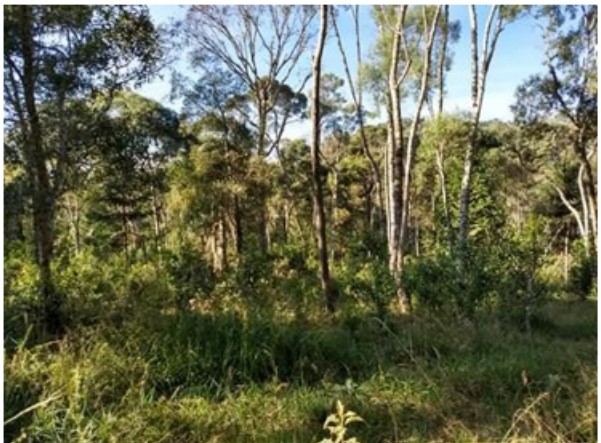

AFS-B (Agroforestry)

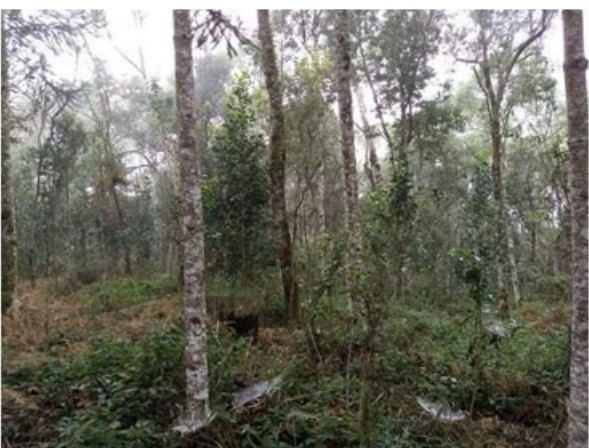

SPS (Silvopastoral System)

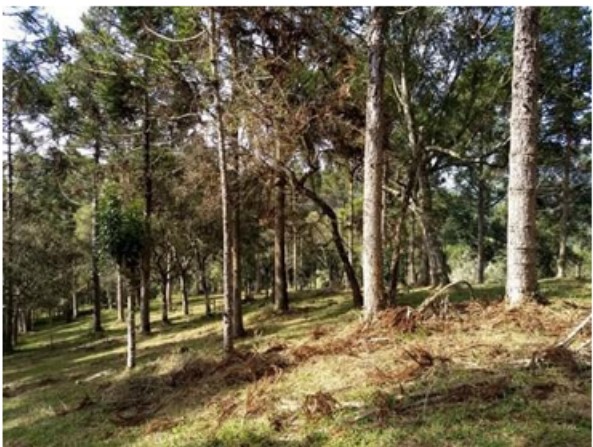

MCS (Monoculture)

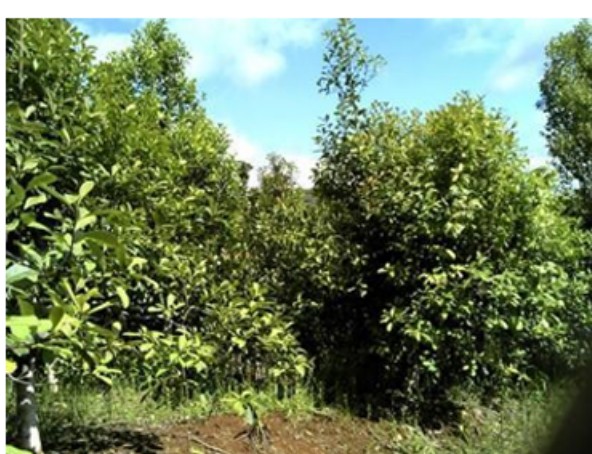

**Figure 1.** General view of the study sites with yerba mate production (according to Table 1).

### 2.2. Analysis of Soil ES Indicators

Seven SES related to soil fertility, nutrient cycling, carbon sequestration, erosion control, health, and biodiversity [23] were chosen, and potential soil quality indicators were defined for each (Table 2).

**Table 2.** Soil ecosystem services (SES) and respective potential soil quality indicators *.

| Soil Ecosystem Services | Description | Soil Quality Indicators |
|---|---|---|
| Soil fertility | Capacity to provide nutrients and produce biomass [42] | pH, potential acidity ($H^+ + Al^{3+}$), cation exchange capacity (CEC), sum of bases (SB) |
| Carbon sequestration | Ability of soils to sequester organic carbon and promote climate mitigation services [10] | Carbon stocks ($C_{stock}$), metabolic quotient ($qCO_2$) |
| Erosion control | Control or prevention of soil loss, provided mainly by vegetation covering the soil [22], | Bulk density (BD), granulometry (Granul), structural stability index (SSI), gravimetric soil water content (GWC) |

**Table 2.** *Cont.*

| Soil Ecosystem Services | Description | Soil Quality Indicators |
|---|---|---|
| Nutrient cycling | Provision of nutrients for plants and to fuel the ensemble of biological processes [42] | Litter production (LitPrd), litter nutrients (LitNut), Beta-glucosidase (Beta-Glu), urease (Ure) activities |
| Plant provision | Soil's ability to store N and meet plant N needs [21] | Total nitrogen (TN), phosphorus (P) |
| Soil biodiversity * | Action of soil organisms that affect ecosystem functions and service provision [43] | Earthworm species richness (EwR), density (EwD) and biomass (EwB) microbial activity (FDA) |
| Soil health * | Integrative property of soil that can be changed by management [44] | Microbial activity: metabolic quotient ($qCO_2$), microbial quotient (qMic), microbial activity (FDA), beta-glucosidase (Beta-Glu) and urease (Ure) activities |

* Potential soil quality indicators related to soil biodiversity and soil health are more complex, e.g., microbial activity also indicates SES related to nutrient cycling, soil fertility, and soil health [45], while earthworms also indicate SES related to the habitat function for invertebrates and microbes, soil water storage and infiltration, soil fertility, and soil health [46].

### 2.3. Analysis of Soil Attributes

Simple soil and litter samples were collected from 10 subplots of 100 m$^2$ (10 m × 10 m), randomly distributed in an area of 2500 m$^2$ (50 m × 50 m), in each production system. Sampling was carried out at the start of the rainy period in November 2018 for soil chemical, physical, and microbiological attributes, and soil macroinvertebrates, and in November 2019 for chemical attributes.

Disturbed soil samples were collected using a straight shovel in mini-trenches (40 cm × 40 cm × 45 cm), in layers of 0–10 cm, 10–20 cm and 20–40 cm, and then sieved and air-dried in the laboratory (<2.0 mm, 40 °C). Simple undisturbed samples were collected with metallic cylinders of 100 cm$^3$ in the middle of each corresponding layer to determine soil bulk density [47].

The pH (CaCl$_2$), potential acidity (H$^+$ + Al$^{3+}$, cmol$_c$ dm$^{-3}$), available cations (Ca$^{2+}$, Mg$^{2+}$, K$^+$, cmol$_c$ dm$^{-3}$), phosphorus (P, mg dm$^{-3}$), the sum of exchangeable bases (SB, cmol$_c$ dm$^{-3}$), aluminium saturation (m, %), base saturation (V, %), and cation exchange capacity (CEC, cmol$_c$ dm$^{-3}$) were assessed following standard methods [47]. Total carbon (TC), nitrogen (TN), and sulphur (S) were determined by dry combustion, according to [48], on a CHNS elemental analyser. The calculation of total carbon (C$_{stock}$) and total nitrogen (N$_{stock}$) soil stocks were performed in each sampled layer (0–10 cm, 10–20 cm and 20–40 cm), using the equation: stocks (Mg ha$^{-1}$) = concentration (g kg$^{-1}$) × layer bulk density (BD, g cm$^{-3}$) × layer thickness (cm) [49]. The total values of C stocks for the soil profile (C$_{stock}$ Mg ha$^{-1}$) to 0–40 cm depth were calculated by adding the values of the respective stocks of the three layers sampled.

The soil structural stability index (SSI), which is used to evaluate the risk of soil structural degradation, considers the organic matter levels required to maintain soil structure [50,51], and was calculated as SSI (%) = {[TC (g kg$^{-1}$) × 1.724]/[silt (g kg$^{-1}$) + clay (g kg$^{-1}$)]} × 100 [52]. Soil moisture was measured based on the gravimetric soil water content (GWC) in volumetric cylinder samples, and soil particle size analysis (granulometry) was conducted using standard methods [47].

The litter accumulated (LitPrd) on the soil surface was sampled in the same subplots using a 0.25 m$^2$ square template [53]. Surface litter chemical analyses included total carbon and nutrients (N, Ca, Mg, K, P and S) and followed the methods outlined in [54], while the carbon, nitrogen, and nutrient stocks were obtained from the concentration of nutrients

and the dry mass of organic material, estimated by the equation: LitNut (Mg ha$^{-1}$) = litter nutrient concentration (g kg$^{-1}$) × dry litter mass (Mg ha$^{-1}$) [55].

Earthworm populations were sampled in the 0–20 cm layer in each of the 10 subplots of the four yerba mate production systems by digging a soil monolith (25 cm × 25 cm to 20 cm depth), following a modification of the standard protocol of the Tropical Soil Biology and Fertility (TSBF) Programme [56], and standardised in the ISO norm 23611-1:2018 [57]. The earthworms were hand sorted from the monoliths in the field and immediately fixed in 92% ethanol for later identification in the laboratory at species level [58,59]. The results were extrapolated per square meter and expressed as earthworm density (EwD, number of individuals m$^{-2}$) and biomass (EwB, g m$^{-2}$ fresh mass). Total species richness (EwR) was obtained considering all the species found in each yerba mate production system.

Soil microbiological analyses were carried out in the 0–10 cm layer in each of the 10 subplots of the four yerba mate production systems. Microbial biomass carbon (C-SMB, mg kg$^{-1}$) was estimated by fumigation–extraction [60]. After moisture correction, microbial-C was calculated from fumigated and non-fumigated samples using a flux conversion factor (Kc) of 0.33 [61]. The soil basal respiration (SBR, µg C-CO$_2$ kg$^{-1}$ h$^{-1}$) was estimated by the amount of CO$_2$ released from the soil during a 7-day incubation period, as described by [62]. The metabolic quotient (qCO$_2$) and microbial quotient (qMic) are the ratios between, respectively, SBR and C-SMB; and C-SMB and total C (TC). The activity of beta-glucosidase (Beta-Glu, mg ρ-nitrophenol kg$^{-1}$ soil h$^{-1}$) and urease (Ure, µg NH$_4$-N g$^{-1}$ soil 2 h$^{-1}$) were determined, respectively, following the methods described in [63,64]. Soil microbial activity was measured by the Fluorescein DiAcetate hydrolysis method (FDA, µg of FDA g$^{-1}$ soil h$^{-1}$) and determined according to [65] modified by [66].

*2.4. Statistical Analysis*

A descriptive statistical analysis of the soil attributes in the sites was performed to verify data dispersion. Means, standard deviations (S) and coefficients of variation (CV) were obtained for each variable.

Principal component analyses (PCA) were performed on the database, and we conducted separate analyses for each layer at site level. Considering all layers, the physical and chemical attributes had *n* = 80, while only the superficial layers had data on microbial activity (*n* = 40) and macroinvertebrates (earthworms) (*n* = 40).

The principal components (PCs) with eigenvalues >1 and variables with a higher loading factor (represented by coordinates $\geq$ 0.65, reflecting higher contribution to the PCs) were assumed to be the variables that best represented system attributes and, therefore, better soil quality indicators [67,68]. On the other hand, the PCs and variables with the eigenvectors associated with lower magnitude were removed from the analysis [69]. A distance-based biplot was used to evaluate and represent the direction and strength of the relationships between the variables and PC scores. The closer the eigenvector is to +1.0 or −1.0, the more important the variable is for the component. Pearson's correlation matrices were performed to determine the strength of relationships among soil indicators. The analyses, which included data scale transformation, covariance matrix computation, eigenvalues and eigenvectors, selection of PC, and correlation matrices, were developed in R [70] using the packages FactoMinerR, Factoshiny, and Corrplot, version 2023.06.1.

*2.5. Analysis of Ecosystem Services Indicators of Soils*

For the qualitative interpretation of the PCA analysis, we selected the soil quality indicators from the 0–10 cm depth with the greatest contributions, which indicated the provision of SES related to soil fertility, carbon sequestration, erosion control, nutrient cycling, plant provision, biodiversity and health, to compare the production systems. Based on [17], the indicators' measured values were normalised on a scale ranging from 0 to 1, as follows:

$$X'_i = \ X_i - \ X_{min} / \ X_{max} - \ X_{min} \tag{1}$$

where $X_i'$ is the normalised (0–1) value, $X_i$ is the measured value, $X_{min}$ and $X_{max}$ are the minimum and the maximum, respectively, of each considered indicator in all samples from the four production systems. Equation (1) gives high priority (i.e., values close to 1) to higher values of the considered indicator (higher is better); the lowest value, 0, does not indicate that the function is not provided, but that it is the lowest at the considered site. If higher values represent soils with lower quality, such as in the case of the indicator BD, we applied Equation (2).

$$X_i' = X_i - X_{max}/ X_{min} - X_{max} \qquad (2)$$

The normalised values of the soil quality indicators were represented graphically on radar plots in order to evaluate the soil multifunctionality expressing the SES by the set of selected indicators, considering the graph area (higher is better) in each yerba mate production system. The relative size of each axis represents the contribution of each indicator of SES on the graph.

## 3. Results and Discussion

### 3.1. Soil Quality Variables in the Production Systems

The soils in the four production systems are clayey to very clayey, with high levels of total carbon, high acidity with low pH (CaCl$_2$) of 4.1 on average in the soil profiles and high exchangeable Al, high potential acidity (H+Al), and aluminium saturation (m). All sites had generally low soil fertility with very low levels of available P, and very low base saturation values (V) and sum of exchangeable bases (SB), which are characteristic of Cambisols from natural areas in the Bituruna region.

Some variables revealed different effects of management, e.g., a residual effect of liming in MCS, in the two topmost layers of the soil, indicated by higher pH values, lower exchangeable Al and H+Al, higher levels of available Ca and Mg, and higher V (%), compared to the other production systems. A residual liming effect was also observed only in the 0–10 cm layer in SPS.

The BD values were typical of uncompacted clayey soils, with AFS-A and AFS-B having the lowest values, and MCS and SPS having the highest values. Nonetheless, especially in SPS, the soil surface layer had the highest BD value, indicating the effect of soil compaction caused by dairy cattle trampling during grazing. In that layer, there were significant differences ($p < 0.05$) in BD values which followed the order AFS-A (0.79 g cm$^{-3}$) $\leq$ MCS (0.89 g cm$^{-3}$) $\approx$ AFS-B (0.92 g cm$^{-3}$) < SPS (1.10 g cm$^{-3}$) [40].

A full descriptive statistical analysis of the soil attributes in the four production systems, and the three soil layers, is provided in Appendix A (Tables A1 and A2). In the following sections, we will briefly present the results of the multivariate analysis (PCAs), the correlations between all variables studied, and the normalised values for the provisioning of ES.

In the principal component analyses (PCA) using only the 0–10 cm data, the first axis explained 39.9% of the total variation of the dataset and was associated mainly with soil chemical variables related to soil fertility (Figure 2A and Appendix A (Table A3). Of these, H+Al and CEC had the largest contributions towards this axis and were opposed to pH and SB. One biochemical (FDA) and one physical soil variable (SSI) were also highly correlated with axis 1. These variables were the ones that most contributed to the differences between the production systems (Figure 2B), indicating that soil fertility was the main driving force in separating the management systems, with the monoculture yerba mate being more associated with variables considered important for soil fertility management (pH and sum of bases), and the agroforestry systems with higher soil physical quality (SSI) and CEC, also associated with soil fertility.

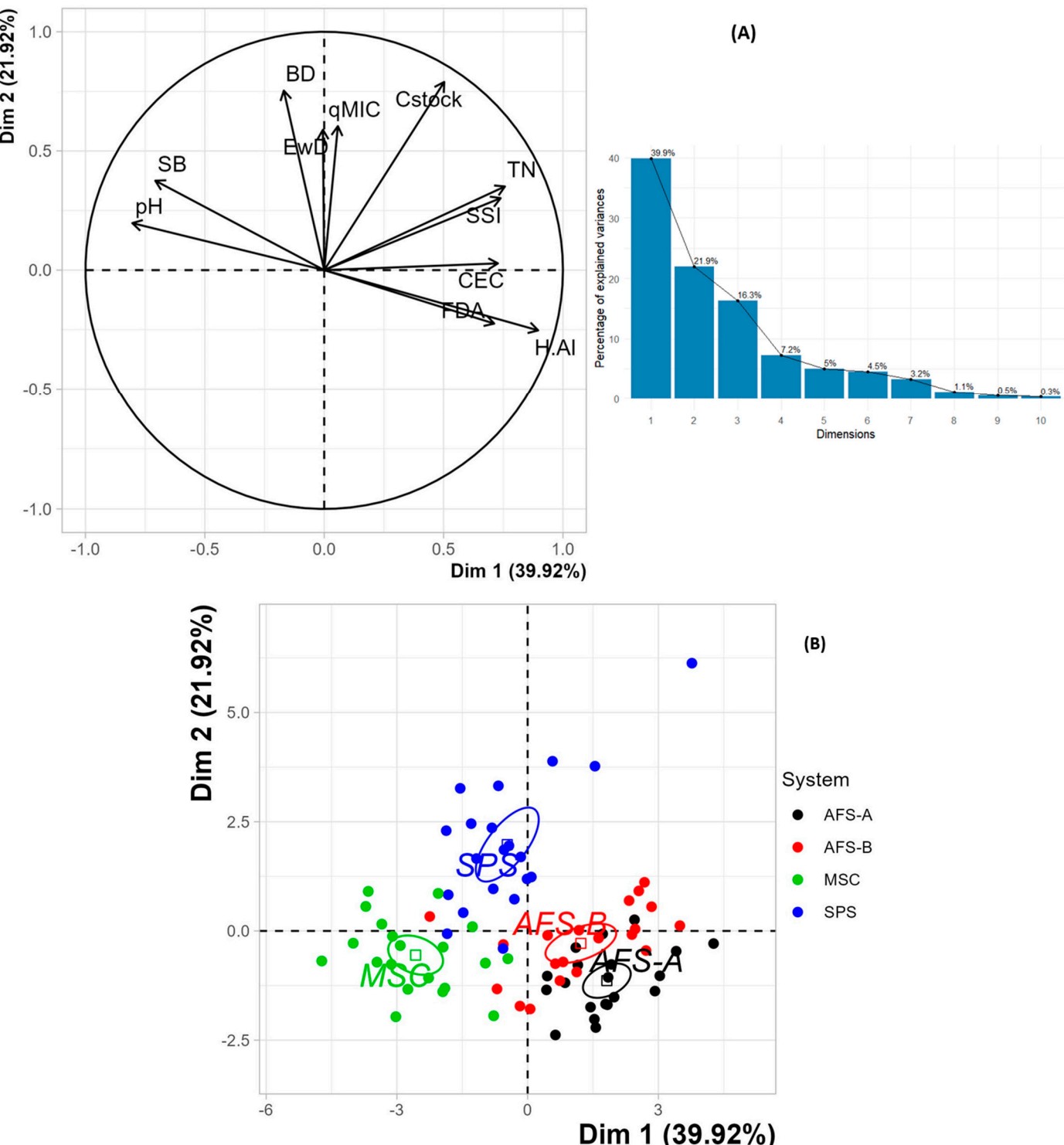

**Figure 2.** (**A**) Correlation circle with projections of studied variables in the 0–10 cm layer on the two main axes (Dim) of a principal component analysis (PCA). The smaller figure shows the eigenvalues of all dimensions. (**B**) PCA showing the distribution of the samples taken in the different production systems.

The second axis explained almost 22% of the total variation in the dataset and was more associated with variables that are affected by soil organic matter contents (Figure 2A), such as qMic, EwD, and $C_{stock}$. Interestingly, soil bulk density (BD) was also highly correlated with axis 2. This axis was also the one that contributed most to separating the SPS from the remaining yerba mate management systems (Figure 2B).

The first two axes of the PCA using the soil variables from the 10–20 cm layer explained 69.4% of the total variation of the dataset (Figure 3A). Once again, the first axis was mainly driven by soil chemical variables that contributed to the separation of the yerba mate management systems (Figure 3B), with 48.5% of the variation explained. The largest contributions were from H+Al, CEC, TN, $C_{stock}$ and available P. The physical soil quality (SSI) was again related to axis 1, and was greater in the agroforestry management systems. However, the data points were more highly dispersed among the land management systems, so the separation was less evident, given that the subsurface layers are less sensitive to management practices.

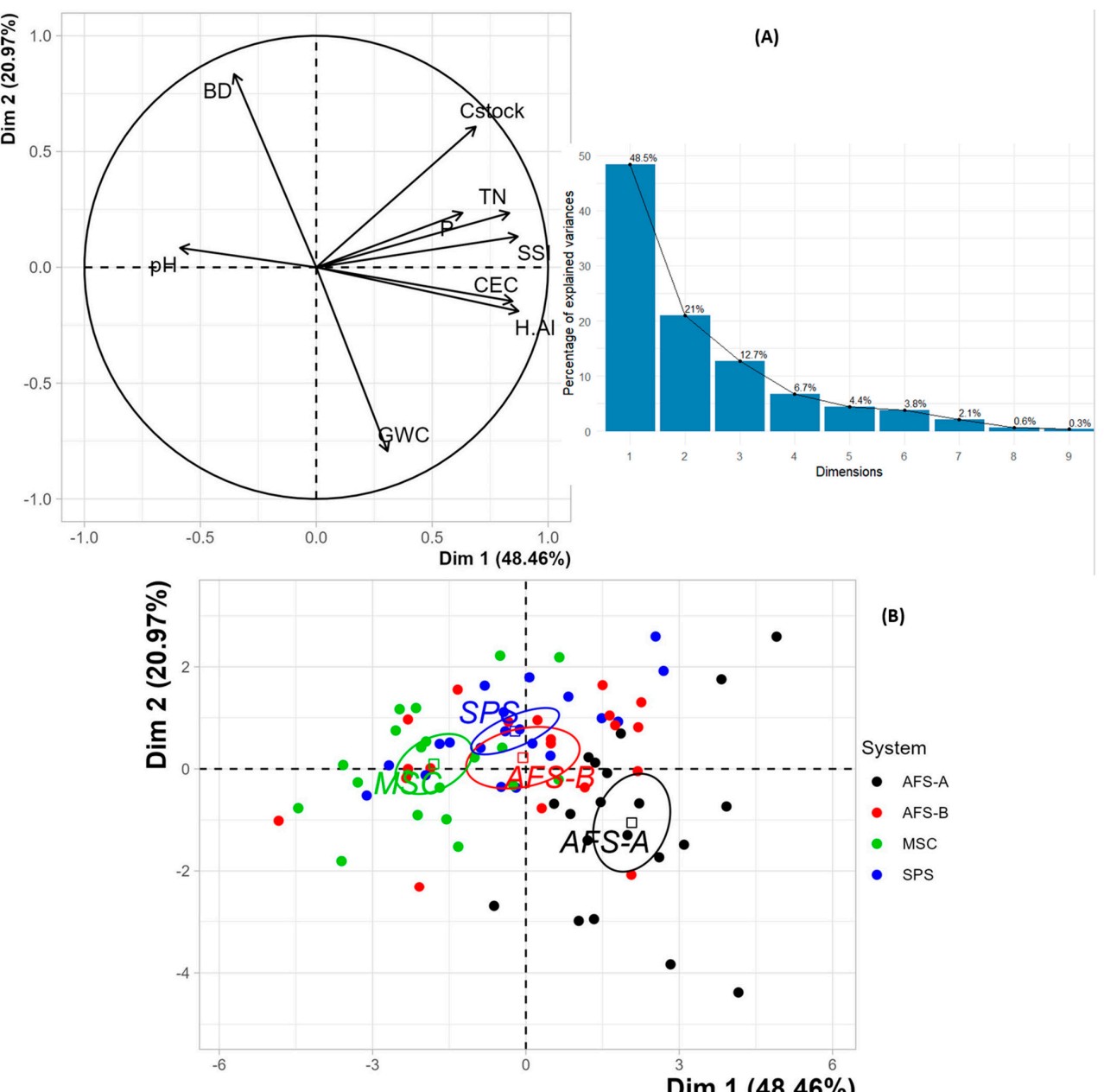

**Figure 3.** (**A**) Correlation circle with projections of studied variables in the 10–20 cm layer on the two main axes (Dim) of a principal component analysis (PCA). The smaller figure shows the eigenvalues of all dimensions. (**B**) PCA showing the distribution of the samples taken in the different production systems.

The second axis explained 21% of the variation in the dataset (Figure 3A) and was associated mainly with GWC and BD. It is interesting to note that GWC separated AFS-A

from the other yerba mate management systems (Figure 3B), probably due to the ability of that system to keep the soil wetter. On the other hand, the effect of soil compaction caused by cattle in SPS can also be seen in the BD values of the 10–20 cm layer.

The first two axes of the PCA using data from the 20–40 cm layer explained 72.7% of the total variation of the dataset (Figure 4A). The first axis explained 48.5% of the variation and was once again related mainly to chemical variables (H+Al, CEC, TN, C$_{stock}$) and the soil physical quality (SSI), being useful to separate the yerba mate management systems (Figure 4B). Axis 2 explained nearly 21% of the total variation and was related mainly with soil particle size distribution (Granul), as opposed to soil moisture contents (GWC), which were useful for separating one of the agroforestry systems (AFS-B) from the remaining yerba mate production systems, due to its lower clay content.

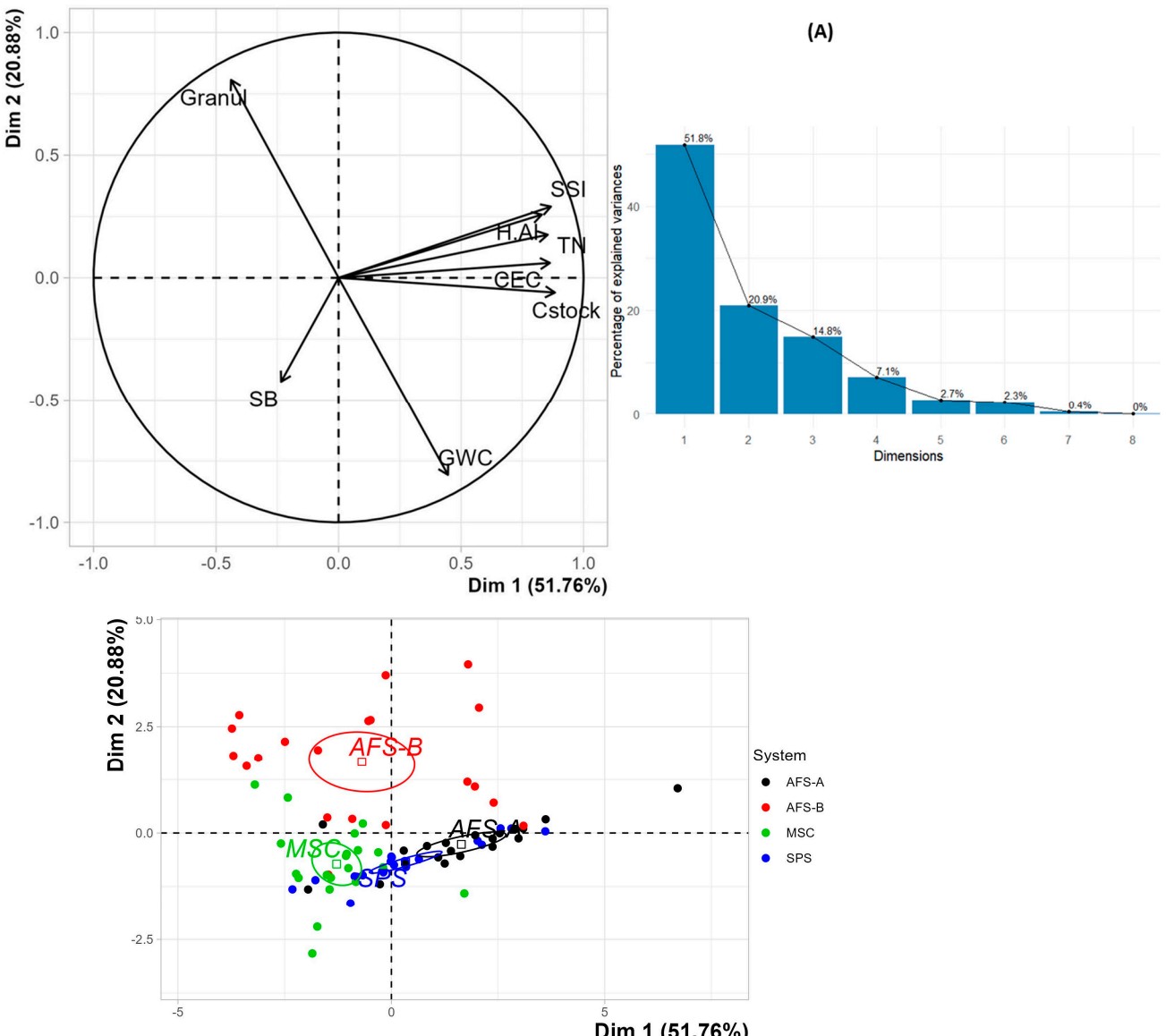

**Figure 4.** (**A**). Correlation circle with projections of studied variables in the 20–40 cm layer on the two main axes (Dim) of a principal component analysis (PCA). The smaller figure shows the eigenvalues of all dimensions. (**B**) PCA showing the distribution of the samples taken in the different production systems.

### 3.2. Correlations among Attributes

The correlations matrix of the data from the 0–10 cm layer showed that qMic was highly correlated with BD and FDA was highly correlated with H+Al, pH and SB (Figure 5

and Appendix A (Table A6). In general, only very weak correlations with other indicators were identified for EwD, with the exception of $C_{stock}$. Soil pH, H+Al, CEC and SB were highly correlated with each other and showed little correlation with qMic. There were strong positive correlations between SSI and $C_{stock}$, CEC and TN, and a negative relationship with BD. Subsurface layers followed the same pattern of correlation between most chemical attributes. The positive correlations between SSI and $C_{stock}$ remained, while negative correlation between GWC and BD remained at 10–20 cm and between GWC and granulometry at 20–40 cm. Soil structure and aggregation are vital physical properties that impact a wide array of SES, including biomass production, water retention and infiltration, erosion control, soil carbon stocks, and biogeochemical cycling of essential elements [71]. SSI had a strong contribution to all the soil layers, and its association with NT, P, CEC, and $C_{stock}$ showed that it is a good indicator of soil quality.

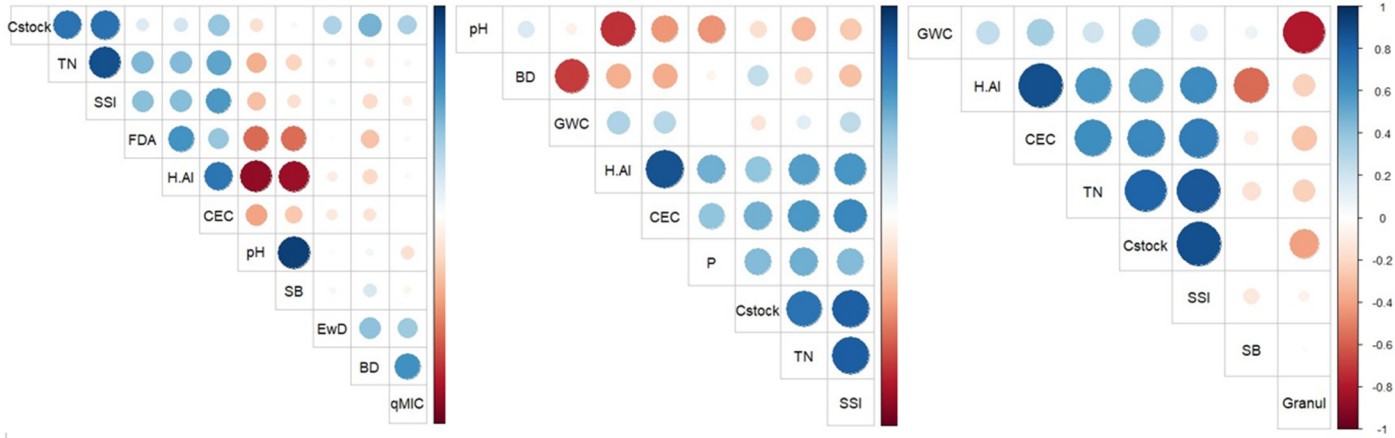

**Figure 5.** Correlation among soil attributes across the soil layers (0–10 cm, 10–20 cm, and 20–40 cm) in the four production systems under study.

### 3.3. Soil Quality Indicators and Soil Ecosystem Services

Soil quality in the different yerba mate production systems was compared using indicators associated with SES delivery. We aimed to answer the following questions: Under different conditions or land uses, what gains or losses of SES provision can be expected? Which land management conditions are best suited to provide a given SES? However, it is important to point out that quantifying SES in areas managed by landowners must involve careful consideration of local management and environmental conditions and variations.

The relationship between soil quality and the provision of SES was explored using radar plots, a practical tool to support farmer management decisions [72–74]. The criteria used to select soil quality indicators take into account several factors, which must be treated in a holistic and complementary way. These soil indicators: (1) should represent key soil processes related to the expression of soil system functions and functioning [75]; (2) can represent an index when integrating two or more soil attributes; (3) are related to SES [23] (Table 2); and (4) are related to soil conditions that are adequate or can favour the development and production of yerba mate.

Soil attributes with greater variation between production systems may reflect the existence of differences between soil and environmental conditions, conservation and soil fertility management practices, yerba mate management and/or agroforestry management, as well as site use history, among others. However, to better serve as indicators of soil quality, these differences between production systems should also represent key soil processes that express functions related to the provision of SES.

The expression of soil quality and health using radar plots depends on selecting a number of soil indicators that together act with multifunctionality, and contribute to the provision of various SES, consequently improving the sustainability of agroforestry

production systems. The larger the area of the radar plot, the better the soil quality and the provision of SES, with the relative values of the axes of each indicator indicating the need for improvements to be made in the management of the production system.

However, it is important to remember that yerba mate evolved naturally in a shaded environment in acidic soils with low levels of available nutrients, and with the participation of symbioses such as mycorrhizal fungi [76], among other microorganisms. Furthermore, there is genetic variability [77,78], and a lack of studies associating yerba mate productive efficiency with nutritional needs and growth parameters, under different plant density and management conditions, as well as shading intensity. Finally, there are few studies on the relationships with soil quality, as well as responses to the application of nutrients from sustainable sources [79]. Hence, soil quality indicators should preferably contribute to improving the development of productive and sustainable management of yerba mate, as well as to ES provision.

The radar plots (Figure 6) included seven indicators (CEC, TN, $C_{stock}$, qMic, FDA, BD and SSI) that made the most significant contributions to the differences between the production systems in PC1 and PC2 (Appendix A—Table A4). These indicators are related to SES provision involving soil fertility, plant production, carbon sequestration, nutrient cycling, soil biodiversity, soil health, and erosion control (Table 2). Although macrofauna communities are good soil indicators that are sensitive to change and disturbance [80], the extreme values found in the present study prevented their use.

Soil microorganisms play an important role in SES provisioning, and the biophysical model showed how FDA and qMic were affected by land use, with higher microbial activity (FDA) in AFS-A and AFS-B, in contrast with low activity in MCS. On the other hand, the higher amount of organic carbon being immobilised in microbial biomass (qMic) in SPS indicates better nutrient cycling associated with this system, possibly related to liming and cattle derived-N.

Similarly, TN concentration in AFS-A and AFS-B was strongly associated with the leaf litter layer, while in SPS, the N was probably from dairy cattle manure and urine added during grazing, as there was less accumulated litter, and no mineral fertiliser was applied. The SSI in both AFS and SPS indicates sufficient soil organic matter to maintain soil structural stability. The lower values in MCS are likely due to low TC contents and in SPS, due to higher BD.

Overall, soil $C_{stock}$ had low contribution as an indicator, highlighting the difficulties associated with using integrative values that depend on multiple variables (in the present case, TC contents and BD).

The production of litter accumulated on the soil surface is an important indicator in agroforestry systems, not only for nutrient cycling but also for soil surface protection [81] AFS-B had the highest surface litter mass (8 Mg ha$^{-1}$), while AFS-A and MCS had around 5 Mg ha$^{-1}$, and SPS had only 2 Mg ha$^{-1}$ (Appendix A—Table A2) [40]. The number and type of trees in yerba mate AFS, including the herbaceous and shrubby strata as well as the organic residues from mowing, are important factors to consider.

Therefore, traditional and agroecological management of yerba mate agroforestry in Araucaria Forest can increase several SES, especially those related to soil health (FDA and qMic), plant provision (TN), and erosion control (SSI).

Hence, the influence of AFS on the provision of SES is in line with the findings of [82], who highlighted the positive effects of AFS on ES in Brazil. These land use systems conserve soil fertility and structure, mitigate climate change, and improve nutrient cycling by creating a favourable shaded environment with continuous input of leaf litter that benefits, for example, soil microbial activity, macrofauna biodiversity, and carbon and nitrogen storage [83–85].

The potential of agroforestry to improve soil properties through fresh organic matter inputs and soil organic carbon increases, thereby increasing the system's capacity to cope with unfavourable edaphoclimatic conditions, is well reported in the literature, e.g., [32,86–88]. In fact, because AFS can support adaptation to climate change by improving ES, these sys-

tems were incorporated into the National Adaptation Plans of several developing countries. The sustainability of agroecosystems depends on their ability to deliver an entire package of multiple ES, rather than provisioning services alone. Hence, new social and ecological dimensions of agricultural management must be explored in agricultural landscapes to foster this ability [15].

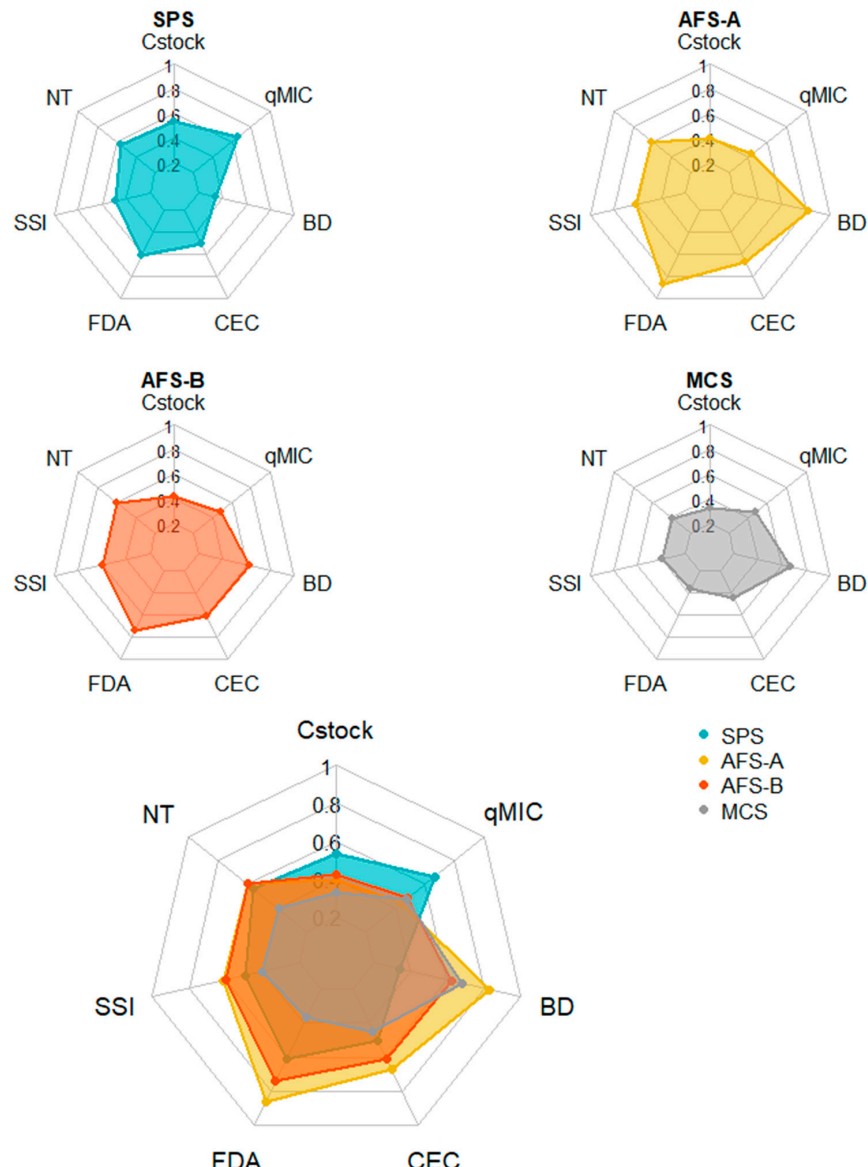

**Figure 6.** Normalised values for the provision of seven ecosystem services indicators in the 0–10 cm layer of four yerba mate production systems. The axes represent normalised relative values of soil quality indicators. Descriptions of the systems can be found in Table 1.

There are many studies on soils and ES but few of them explore the direct relationship with soil properties [10]. In this paper, a range of SES were identified, requiring quantification of soil quality indicators, most of them widely used in standard or commercial soil analyses. However, besides quantification, the need for greater and widespread awareness of soil functions and services in landscape planning is evident. The work we undertook revealed the challenges connected to the selection of suitable and widely applicable indicators. To achieve this, further efforts are required to raise awareness of the importance of soils in a landscape context, especially among planners and local decision-makers.

Furthermore, integrated assessment of multiple SES to support land use planning at the landscape level is necessary for sustainable use of natural resources, to identify possible

trade-offs regarding SES provision, and to balance the needs of different stakeholder groups [89]. Such approaches for ES at a landscape level provide a simplified and rapid method for evaluating the impact of land use and land management options on ES.

In this study, we demonstrated the utility of an indicator-based approach by applying it to a set of soil attributes in four production systems based on yerba mate. From a set of soil function indicators as a requirement for evaluating ES provision as a function of soil management, the indicator-based approach, e.g., [17], focuses on exploring aspects of soil properties and diversity of soil biota [10] and methodological frameworks [13] and represents an advance toward the of assessment of SES from available soil data. This provided key insights into the functioning of agroecosystems which, combined with agroecological practices, maximises ES provision. Since this approach allows assessment of the risk of loss, maintenance, or improvement of SES provision, the results are also useful for supporting policy and management decisions related to the yerba mate production systems.

From the approach using the most sensitive indicators, the challenge is to quantify SES in monetary terms [90], addressing the issue with a more economic focus. Although the idea of value is anthropocentric and can only grasp a limited aspect of the whole value of an ecosystem or service [23], this approach can shed light on the role of economic valuation of SES, allowing such values to be made explicit to society in general and policymakers in particular [43].

## 4. Conclusions

ES-based management is a promising way of ensuring the sustainability of production systems. The methods used in this study allowed the analyses of trade-offs of SES in land uses, including four yerba mate production systems in southern Brazil. Several soil quality indicators explained the differences among the four production systems at various depths. At the surface layer, the indicators were CEC, TN, $C_{stock}$, qMic, FDA, BD, and SSI. In the context of this study, when yerba mate was planted among native yerba mate and secondary forest (AFS), higher SES provisioning was observed compared with that of the other systems.

The indicator-based approach used was able to capture the soil functions and offered a good starting point for quantifying SES provision. Therefore, it can be used as a tool for decision-makers to evaluate land-use change-based impacts on SES, highlighting more sustainable management practices.

Furthermore, the methods used can be expanded to include additional soil quality indicators and services, and these results can be combined with additional studies to analyse trade-offs in SES. Visualising and graphically representing observed and modelled data is necessary to improve the science–policy interface. Future improvements are also needed, such as the inclusion of other ES and other sites, soil types, and production systems to validate this methodology.

**Author Contributions:** Conceptualization, L.M.P. and R.T.d.G.P.; methodology, L.M.P., R.T.d.G.P., K.d.S. and G.G.B.; software, L.M.P.; validation, L.M.P., R.T.d.G.P., K.d.S. and G.G.B.; formal analysis, L.M.P., R.T.d.G.P., K.d.S. and G.G.B.; investigation, L.M.P., R.T.d.G.P., K.d.S. and G.G.B.; resources, L.M.P.; data curation, L.M.P., R.T.d.G.P., K.d.S. and G.G.B.; writing—original draft preparation, L.M.P.; writing—review and editing, L.M.P., R.T.d.G.P. and G.G.B.; visualization, L.M.P., R.T.d.G.P. and G.G.B.; supervision, L.M.P.; project administration, L.M.P.; funding acquisition, L.M.P. and G.G.B. All authors have read and agreed to the published version of the manuscript.

**Funding:** This research was funded by Brazilian Agricultural Research Corporation (Embrapa, grant number 16.16.05.002.00.01), and Brazilian National Council for Scientific and Technological Development (CNPq, grant number 310690/2017-0, 441930/2020-4 and 312824/2022-0).

**Data Availability Statement:** Data can be made available on request.

**Acknowledgments:** The authors acknowledge the farmers Anísio Francisco da Rosa, Bernardo Vergopolen, Miguel Corrêa, and Rosildo Stangherlin for making their farms available for this study. Additionally, we acknowledge Jonatas Gueller, Mario Kioshi Yamada, and Marcos Gonçalves Tenorio

for their assistance with the data collection and Osmir Lavoranti for his tips regarding the statistical analysis.

**Conflicts of Interest:** The authors declare no conflict of interest.

## Appendix A

**Table A1.** Descriptive statistics of chemical and physical soil attributes at different depths in sites with yerba mate production systems (Paraná, Brazil).

| System | Layer | Statistics | pH | Al | H+Al | Ca | Mg | K | SB | CEC | TN | TC | P | $C_{stock}$ | Sand | Silt | Clay | SSI | GWC | BD |
|---|---|---|---|---|---|---|---|---|---|---|---|---|---|---|---|---|---|---|---|---|
| SPS | 0–10 cm | Mean | 4.3 | 1.4 | 11.4 | 4.8 | 1.2 | 0.2 | 6.2 | 18.8 | 0.42 | 46.84 | 2.4 | 51.4 | 36.6 | 297.1 | 666.4 | 8.4 | 0.3 | 1.1 |
| | | S | 0.3 | 1.1 | 2.7 | 2.4 | 0.9 | 0.0 | 3.0 | 3.6 | 0.09 | 13.12 | 0.8 | 14.3 | 5.6 | 41.0 | 42.5 | 2.3 | 0.0 | 0.1 |
| | | CV (%) | 6.8 | 76.1 | 23.8 | 49.2 | 70.5 | 24.4 | 48.6 | 19.1 | 21.49 | 28.0 | 34.5 | 27.7 | 15.2 | 13.8 | 6.4 | 27.9 | 6.3 | 5.5 |
| | 10–20 cm | Mean | 3.9 | 3.5 | 14.8 | 1.0 | 0.3 | 0.1 | 1.5 | 18.9 | 0.30 | 32.67 | 1.2 | 33.9 | 36.5 | 277.9 | 685.6 | 5.8 | 0.3 | 1.0 |
| | | S | 0.1 | 0.9 | 2.7 | 0.9 | 0.2 | 0.0 | 1.1 | 3.2 | 0.06 | 7.56 | 0.5 | 7.5 | 15.1 | 49.2 | 50.6 | 1.4 | 0.1 | 0.1 |
| | | CV (%) | 2.4 | 24.8 | 18.3 | 85.3 | 69.3 | 24.7 | 73.4 | 17.1 | 19.51 | 23.1 | 42.6 | 22.0 | 41.3 | 17.7 | 7.4 | 23.6 | 19.4 | 6.1 |
| | 20–40 cm | Mean | 4.0 | 3.8 | 13.7 | 0.5 | 0.1 | 0.1 | 0.7 | 17.6 | 0.24 | 27.39 | 0.7 | 58.5 | 29.0 | 252.6 | 718.4 | 4.9 | 0.4 | 1.1 |
| | | S | 0.2 | 0.7 | 1.9 | 0.7 | 0.1 | 0.0 | 0.8 | 3.9 | 0.05 | 6.37 | 0.2 | 14.6 | 3.1 | 44.1 | 43.1 | 1.1 | 0.0 | 0.0 |
| | | CV (%) | 6.1 | 19.1 | 14.2 | 130.2 | 103.5 | 27.8 | 111.6 | 22.1 | 20.14 | 23.3 | 35.1 | 25.0 | 10.7 | 17.4 | 6.0 | 23.1 | 8.6 | 3.6 |
| AFS-A | 0–10 cm | Mean | 3.7 | 5.0 | 19.2 | 0.6 | 0.5 | 0.2 | 1.3 | 18.0 | 0.44 | 52.17 | 2.1 | 41.3 | 64.5 | 305.5 | 630.0 | 9.6 | 0.4 | 0.8 |
| | | S | 0.2 | 1.0 | 3.9 | 0.6 | 0.3 | 0.0 | 0.9 | 1.6 | 0.06 | 7.50 | 0.5 | 6.9 | 22.4 | 23.9 | 20.9 | 1.4 | 0.1 | 0.1 |
| | | CV (%) | 4.4 | 18.9 | 20.6 | 100.1 | 65.7 | 24.0 | 70.2 | 9.1 | 13.47 | 14.4 | 25.0 | 16.7 | 34.7 | 7.8 | 3.3 | 14.8 | 22.8 | 10.1 |
| | 10–20 cm | Mean | 3.8 | 5.4 | 19.5 | 0.3 | 0.3 | 0.1 | 0.7 | 16.9 | 0.33 | 41.25 | 1.2 | 35.7 | 64.3 | 292.7 | 643.0 | 7.6 | 0.5 | 0.9 |
| | | S | 0.1 | 0.8 | 3.0 | 0.3 | 0.2 | 0.0 | 0.4 | 2.6 | 0.05 | 6.03 | 0.5 | 7.8 | 22.8 | 22.1 | 18.2 | 1.1 | 0.2 | 0.1 |
| | | CV (%) | 2.6 | 14.5 | 15.3 | 76.3 | 79.2 | 19.4 | 60.9 | 15.6 | 16.28 | 14.6 | 43.7 | 21.9 | 35.5 | 7.6 | 2.8 | 14.3 | 31.1 | 11.4 |
| | 20–40 cm | Mean | 3.9 | 4.8 | 17.6 | 0.2 | 0.1 | 0.1 | 0.4 | 21.1 | 0.25 | 31.48 | 0.5 | 61.8 | 58.2 | 271.6 | 670.2 | 5.7 | 0.5 | 1.0 |
| | | S | 0.1 | 0.6 | 2.8 | 0.1 | 0.1 | 0.0 | 0.2 | 2.8 | 0.06 | 6.08 | 0.3 | 11.5 | 39.9 | 20.4 | 43.8 | 1.0 | 0.0 | 0.1 |
| | | CV (%) | 2.3 | 12.6 | 16.1 | 52.7 | 92.9 | 70.9 | 45.7 | 13.4 | 24.65 | 19.3 | 55.2 | 18.7 | 68.5 | 7.5 | 6.5 | 18.1 | 7.6 | 6.8 |
| AFS-B | 0–10 cm | Mean | 3.8 | 3.7 | 17.6 | 1.0 | 0.6 | 0.2 | 1.8 | 15.3 | 0.44 | 46.54 | 2.2 | 43.0 | 149.7 | 350.3 | 500.0 | 9.4 | 0.4 | 0.9 |
| | | S | 0.3 | 1.5 | 4.1 | 1.3 | 0.4 | 0.1 | 1.6 | 5.0 | 0.08 | 8.54 | 0.7 | 8.9 | 46.9 | 18.0 | 54.1 | 1.7 | 0.1 | 0.1 |
| | | CV (%) | 8.1 | 40.2 | 23.2 | 130.9 | 66.3 | 62.1 | 89.9 | 32.9 | 17.60 | 18.4 | 30.1 | 20.7 | 31.4 | 5.1 | 10.8 | 17.8 | 16.3 | 9.6 |
| | 10–20 cm | Mean | 3.9 | 3.7 | 15.1 | 0.4 | 0.2 | 0.1 | 0.7 | 14.0 | 0.30 | 31.66 | 1.0 | 31.5 | 151.2 | 354.0 | 494.8 | 6.4 | 0.4 | 1.0 |
| | | S | 0.3 | 1.5 | 5.0 | 0.3 | 0.1 | 0.0 | 0.4 | 3.1 | 0.05 | 5.75 | 0.3 | 6.8 | 52.1 | 20.4 | 69.0 | 1.0 | 0.1 | 0.1 |
| | | CV (%) | 7.8 | 40.9 | 33.2 | 92.4 | 63.8 | 33.0 | 60.3 | 21.8 | 17.13 | 18.2 | 32.8 | 21.7 | 34.4 | 5.7 | 13.9 | 16.2 | 18.8 | 8.2 |
| | 20–40 cm | Mean | 4.0 | 3.3 | 13.5 | 0.3 | 0.2 | 0.1 | 0.5 | 16.8 | 0.24 | 25.44 | 0.7 | 50.7 | 153.8 | 345.8 | 500.4 | 5.2 | 0.4 | 1.0 |
| | | S | 0.2 | 1.4 | 4.5 | 0.2 | 0.1 | 0.0 | 0.2 | 3.2 | 0.05 | 5.39 | 0.7 | 10.5 | 56.4 | 26.1 | 73.3 | 1.0 | 0.1 | 0.0 |
| | | CV (%) | 4.2 | 43.7 | 33.1 | 72.0 | 69.5 | 33.6 | 48.9 | 19.3 | 20.36 | 21.2 | 88.4 | 20.8 | 36.7 | 7.5 | 14.7 | 19.8 | 21.0 | 2.1 |
| MSC | 0–10 cm | Mean | 4.8 | 0.8 | 8.1 | 5.5 | 2.8 | 0.2 | 8.5 | 14.5 | 0.33 | 39.71 | 1.4 | 35.2 | 70.7 | 351.3 | 578.0 | 7.4 | 0.4 | 0.9 |
| | | S | 0.7 | 1.1 | 4.6 | 3.6 | 2.8 | 0.1 | 4.9 | 2.4 | 0.06 | 5.99 | 0.7 | 5.0 | 27.0 | 49.9 | 73.8 | 1.3 | 0.2 | 0.1 |
| | | CV (%) | 15.3 | 145.7 | 56.5 | 66.2 | 100.3 | 40.8 | 58.2 | 16.8 | 17.91 | 15.1 | 45.9 | 14.2 | 38.2 | 14.2 | 12.8 | 17.4 | 45.5 | 10.1 |
| | 10–20 cm | Mean | 4.3 | 1.7 | 10.0 | 3.4 | 1.0 | 0.1 | 4.5 | 13.8 | 0.25 | 29.98 | 0.7 | 30.2 | 82.6 | 323.2 | 594.2 | 5.6 | 0.4 | 1.0 |
| | | S | 0.5 | 1.4 | 4.4 | 3.2 | 1.0 | 0.0 | 3.9 | 3.8 | 0.04 | 5.20 | 0.4 | 6.4 | 39.3 | 36.5 | 62.7 | 0.9 | 0.1 | 0.1 |
| | | CV (%) | 12.6 | 82.0 | 44.2 | 94.2 | 95.1 | 26.1 | 86.4 | 27.8 | 16.87 | 17.3 | 47.5 | 21.3 | 47.5 | 11.3 | 10.5 | 15.9 | 16.0 | 7.8 |

**Table A1.** *Cont.*

| System | Layer | Statistics | pH | Al | H+Al | Ca | Mg | K | SB | CEC | TN | TC | P | $C_{stock}$ | Sand | Silt | Clay | SSI | GWC | BD |
|---|---|---|---|---|---|---|---|---|---|---|---|---|---|---|---|---|---|---|---|---|
| | 20–40 cm | Mean | 4.3 | 2.0 | 10.4 | 2.1 | 0.6 | 0.1 | 2.8 | 12.2 | 0.20 | 23.24 | 0.5 | 52.7 | 76.6 | 294.0 | 629.4 | 4.3 | 0.4 | 1.1 |
| | | S | 0.4 | 1.1 | 2.6 | 2.4 | 0.8 | 0.0 | 3.0 | 2.1 | 0.04 | 4.11 | 0.3 | 9.0 | 38.2 | 31.2 | 65.6 | 0.7 | 0.0 | 0.1 |
| | | CV (%) | 9.5 | 56.2 | 25.2 | 112.7 | 136.9 | 28.6 | 107.9 | 17.6 | 18.44 | 17.7 | 52.7 | 17.1 | 49.8 | 10.6 | 10.4 | 16.1 | 11.3 | 6.2 |

Where S = standard deviation; CV = coefficient of variation. H+Al, $Ca^{+2}$, $Mg^{+2}$, $K^{+1}$, CEC ($cmol_c$ $dm^{-3}$), SB, V, m (%), P (mg $dm^{-3}$), TOC, sand, silt, clay (g $kg^{-1}$), SSI (%), $C_{stock}$, $N_{stock}$ (Mg $ha^{-1}$), GWC (kg $kg^{-1}$), BD (g $cm^{-3}$).

**Table A2.** Descriptive statistics of microbiological soil attributes at 0–10 cm, earthworms at 0–20 cm layers, and litter production and nutrition data in sites with yerba mate production systems (Paraná, Brazil).

| System | Statistics | C-SMB | SBR | $qCO_2$ | qMic | Beta-Glu | Ure | FDA | EwR | EwD | EwB | $Lit_{Nut}$ | $Lit_{Prd}$ |
|---|---|---|---|---|---|---|---|---|---|---|---|---|---|
| SPS | Mean | 876.9 | 1.4 | 1.6 | 2.0 | 161.3 | 136.6 | 7.0 | 0.7 | 20.8 | 2.3 | 82.2 | 2.4 |
| | S | 69.2 | 0.3 | 0.4 | 0.4 | 29.5 | 48.9 | 0.9 | 0.8 | 24.4 | 5.7 | 40.2 | 1.0 |
| | CV (%) | 7.9 | 20.3 | 23.7 | 23.0 | 18.3 | 35.8 | 12.8 | 114.5 | 117.3 | 245.5 | 48.9 | 42.7 |
| AFS-A | Mean | 669.4 | 1.7 | 2.6 | 1.3 | 243.5 | 148.7 | 9.2 | 0.3 | 4.8 | 26.4 | 140.4 | 5.0 |
| | S | 115.6 | 0.3 | 0.5 | 0.2 | 58.8 | 8.5 | 0.8 | 0.7 | 10.5 | 55.6 | 37.9 | 1.4 |
| | CV (%) | 17.3 | 15.9 | 19.9 | 16.3 | 24.1 | 5.7 | 9.0 | 219.0 | 219.0 | 210.6 | 27.0 | 29.1 |
| AFS-B | Mean | 601.1 | 1.2 | 2.1 | 1.4 | 181.9 | 146.2 | 8.1 | 0.3 | 6.4 | 7.4 | 264.3 | 7.8 |
| | S | 86.1 | 0.3 | 0.5 | 0.3 | 35.8 | 23.3 | 0.8 | 0.5 | 10.9 | 20.1 | 61.7 | 2.3 |
| | CV (%) | 14.3 | 22.8 | 22.2 | 18.0 | 19.7 | 15.9 | 9.5 | 156.7 | 170.1 | 271.8 | 23.3 | 29.0 |
| MSC | Mean | 452.9 | 1.1 | 2.4 | 1.2 | 155.8 | 85.3 | 5.0 | 0.0 | 0.0 | 0.0 | 214.7 | 5.7 |
| | S | 109.1 | 0.3 | 1.0 | 0.3 | 40.6 | 23.0 | 1.3 | 0.0 | 0.0 | 0.0 | 106.4 | 2.1 |
| | CV (%) | 24.1 | 32.5 | 39.0 | 28.4 | 26.1 | 27.0 | 25.1 | 0.0 | 0.0 | 0.0 | 49.6 | 37.0 |

Where C-SMB (mg $kg^{-1}$), SBR ($\mu$g C-$CO_2$ $kg^{-1}$ $h^{-1}$), $qCO_2$ = RBS/C-SMB, qMic = C-SMB/ total organic C, Beta-Glu (mg $\rho$-nitrophenol $kg^{-1}$ soil $h^{-1}$), Ure ($\mu$g $NH_4$-N $g^{-1}$ soil 2 $h^{-1}$), FDA ($\mu$g FDA $g^{-1}$ soil $h^{-1}$), EwR (mean species richness), EwD (individuals $m^{-2}$), EwB (g $m^{-2}$), $Lit_{Prd}$ and $Lit_{Nut}$ (Mg.$ha^{-1}$).

**Table A3.** Load matrix and variance explained by principal component analysis based on the soil attributes dataset for the layers.

| Layer | Eigenvalues | PC1 * | PC2 | PC3 | PC4 | PC5 | PC6 | PC7 | PC8 | PC9 | PC10 | PC11 |
|---|---|---|---|---|---|---|---|---|---|---|---|---|
| 0–10 cm | Variance | 4.391 | 2.412 | 1.797 | 0.796 | 0.551 | 0.496 | 0.357 | 0.120 | 0.051 | 0.028 | 0.000 |
| | % of variance | 39.921 | 21.924 | 16.337 | 7.233 | 5.013 | 4.506 | 3.249 | 1.092 | 0.466 | 0.259 | 0.000 |
| | Cumulative % of variance | 39.921 | 61.844 | 78.181 | 85.415 | 90.427 | 94.934 | 98.183 | 99.275 | 99.741 | 100.000 | 100.000 |
| 10–20 cm | Variance | 4.362 | 1.887 | 1.143 | 0.605 | 0.394 | 0.339 | 0.185 | 0.057 | 0.028 | | |
| | % of variance | 48.462 | 20.969 | 12.699 | 6.726 | 4.372 | 3.765 | 2.055 | 0.636 | 0.316 | | |
| | Cumulative % of variance | 48.462 | 69.431 | 82.130 | 88.856 | 93.228 | 96.994 | 99.049 | 99.684 | 100.000 | | |
| 20–40 cm | Variance | 4.141 | 1.671 | 1.186 | 0.564 | 0.217 | 0.187 | 0.034 | 0.000 | | | |
| | % of variance | 51.764 | 20.881 | 14.829 | 7.056 | 2.706 | 2.333 | 0.429 | 0.000 | | | |
| | Cumulative % of variance | 51.764 | 72.646 | 87.475 | 94.531 | 97.237 | 99.571 | 100.000 | 100.000 | | | |

Where PC * = principal component. Eigenvalues represent the total amount of variance that can be explained by a given principal component.

**Table A4.** PCA representing PCs by soil attributes along with factor loadings of two factors after varimax rotation for the dataset. Bold numbers: relevant factors. Loadings (>0.65).

| | Eigenvalues | Coord | | ctr | | Cos2 | |
|---|---|---|---|---|---|---|---|
| Layer | Variables | PC1 | PC2 | PC1 | PC2 | PC1 | PC2 |
| 0–10 cm | pH | **−0.80** | 0.20 | **14.71** | 1.61 | 0.65 | 0.04 |
| | H.Al | **0.90** | −0.25 | **18.30** | 2.65 | 0.80 | 0.06 |
| | TN | **0.76** | 0.35 | **13.05** | 5.11 | 0.57 | 0.12 |
| | SB | **−0.71** | 0.38 | **11.40** | 5.84 | 0.50 | 0.14 |
| | CEC | **0.73** | 0.03 | **12.06** | 0.03 | 0.53 | 0.00 |
| | $C_{stock}$ | 0.50 | **0.79** | 5.76 | **25.84** | 0.25 | 0.62 |
| | BD | −0.17 | **0.75** | 0.65 | **23.52** | 0.03 | 0.57 |
| | qMic | 0.06 | 0.60 | 0.07 | 15.14 | 0.00 | 0.37 |
| | FDA | **0.71** | −0.22 | **11.55** | 2.09 | 0.51 | 0.05 |
| | EwD | −0.01 | 0.59 | 0.00 | 14.38 | 0.00 | 0.35 |
| | SSI | **0.74** | 0.30 | **12.44** | 3.79 | 0.55 | 0.09 |
| 10–20 cm | pH | −0.59 | 0.08 | 7.91 | 0.37 | 0.34 | 0.01 |
| | H.Al | **0.87** | −0.19 | **17.44** | 1.89 | 0.76 | 0.04 |
| | TN | **0.83** | 0.23 | **15.90** | 2.92 | 0.69 | 0.06 |
| | CEC | **0.85** | −0.15 | **16.43** | 1.13 | 0.72 | 0.02 |
| | P | 0.63 | 0.24 | 9.12 | 2.94 | 0.40 | 0.06 |
| | $C_{stock}$ | **0.69** | 0.61 | **10.84** | 19.55 | 0.47 | 0.37 |
| | GWC | 0.31 | **−0.79** | 2.17 | **33.37** | 0.09 | 0.63 |
| | BD | −0.35 | **0.83** | 2.88 | **36.89** | 0.13 | 0.70 |
| | SSI | **0.87** | 0.13 | **17.32** | 0.94 | 0.76 | 0.02 |
| 20–40 cm | H.Al | **0.83** | 0.26 | 16.62 | 3.98 | 0.69 | 0.07 |
| | TN | **0.85** | 0.18 | **17.60** | 1.85 | 0.73 | 0.03 |
| | SB | −0.23 | −0.42 | 1.33 | 10.77 | 0.05 | 0.18 |
| | CEC | **0.86** | 0.06 | **17.98** | 0.22 | 0.74 | 0.00 |
| | $C_{stock}$ | **0.88** | −0.06 | **18.84** | 0.22 | 0.78 | 0.00 |
| | GWC | 0.45 | **−0.81** | 4.82 | 38.96 | 0.20 | 0.65 |
| | SSI | **0.87** | 0.29 | **18.15** | 5.05 | 0.75 | 0.08 |
| | Granul | −0.44 | **0.81** | 4.65 | **38.97** | 0.19 | 0.65 |

Where coord = loadings of variables that give the coordinates of the variables, normed to 1; ctr = contribution of a variable to a given PC (%) = (var.cos2 * 100)/(total cos2 of the PC); cos2 = represents the quality of representation for variables on the factor map; it is calculated as the squared coordinates: var.cos2 = var.coord * var.coord.

**Table A5.** PCA representing PCs by land uses (supplementary categories) along with factor loadings after varimax rotation for the layers.

| | Eigenvalues | Coord | | V.test | |
|---|---|---|---|---|---|
| Layer | System | PC1 | PC2 | PC1 | PC2 |
| 0–10 cm | SPS | −0.47 | 1.98 | −1.15 | 6.53 |
| | MCS | −2.57 | −0.55 | −6.29 | −1.83 |
| | AFS-A | 1.82 | −1.14 | 4.45 | −3.76 |
| | AFS-B | 1.22 | −0.28 | 2.99 | −0.94 |

**Table A5.** *Cont.*

| | | Eigenvalues | | Coord | | V.test | |
|---|---|---|---|---|---|---|---|
| **Layer** | **System** | **PC1** | **PC2** | **PC1** | **PC2** | | |
| 10–20 cm | SPS | −0.21 | 0.74 | −0.52 | 2.76 | | |
| | MCS | −1.80 | 0.10 | −4.42 | 0.36 | | |
| | AFS-A | 2.07 | −1.05 | 5.09 | −3.94 | | |
| | AFS-B | −0.06 | 0.22 | −0.14 | 0.82 | | |
| 20–40 cm | SPS | 0.33 | −0.68 | 0.84 | −2.71 | | |
| | MCS | −1.28 | −0.72 | −3.23 | −2.88 | | |
| | AFS-A | 1.64 | −0.26 | 4.13 | −1.05 | | |
| | AFS-B | −0.69 | 1.67 | −1.74 | 6.63 | | |

Where V-Test expresses, in number of standard deviations, the difference between the average $m_k$ of group k, and the overall average m: v-test = $m_k - m/s_k$.

**Table A6.** Correlation coefficients of the soil attributes selected as indicators based on Pearson correlation analysis for each of the depth layers.

| 0–10 cm variable | pH | H+Al | TN | SB | CEC | $C_{stock}$ | BD | qMic | FDA | EwD | SSI |
|---|---|---|---|---|---|---|---|---|---|---|---|
| pH | 100.000 | −0.880 | −0.359 | 0.933 | −0.398 | −0.163 | 0.056 | −0.157 | −0.560 | −0.023 | −0.293 |
| H+Al | −0.880 | 100.000 | 0.437 | −0.855 | 0.725 | 0.186 | −0.192 | 0.039 | 0.600 | −0.095 | 0.423 |
| TN | −0.359 | 0.437 | 100.000 | −0.215 | 0.528 | 0.735 | −0.079 | 0.042 | 0.443 | 0.051 | 0.879 |
| SB | 0.933 | −0.855 | −0.215 | 100.000 | −0.262 | 0.032 | 0.161 | −0.060 | −0.552 | 0.047 | −0.164 |
| CEC | −0.398 | 0.725 | 0.528 | −0.262 | 100.000 | 0.389 | −0.142 | −0.008 | 0.382 | −0.115 | 0.571 |
| $C_{stock}$ | −0.163 | 0.186 | 0.735 | 0.032 | 0.389 | 100.000 | 0.462 | 0.320 | 0.142 | 0.303 | 0.744 |
| BD | 0.056 | −0.192 | −0.079 | 0.161 | −0.142 | 0.462 | 100.000 | 0.604 | −0.287 | 0.405 | −0.190 |
| qMic | −0.157 | 0.039 | 0.042 | −0.060 | −0.008 | 0.320 | 0.604 | 100.000 | −0.039 | 0.358 | −0.085 |
| FDA | −0.560 | 0.600 | 0.443 | −0.552 | 0.382 | 0.142 | −0.287 | −0.039 | 100.000 | −0.019 | 0.413 |
| EwD | −0.023 | −0.095 | 0.051 | 0.047 | −0.115 | 0.303 | 0.405 | 0.358 | −0.019 | 100.000 | 0.048 |
| SSI | −0.293 | 0.423 | 0.879 | −0.164 | 0.571 | 0.744 | −0.190 | −0.085 | 0.413 | 0.048 | 100.000 |

| 10–20 cm variable | pH | H+Al | TN | CEC | P | $C_{stock}$ | GWC | BD | SSI |
|---|---|---|---|---|---|---|---|---|---|
| pH | 100.000 | −0.721 | −0.325 | −0.437 | −0.445 | −0.168 | −0.076 | 0.158 | −0.264 |
| H+Al | −0.721 | 100.000 | 0.558 | 0.868 | 0.490 | 0.392 | 0.309 | −0.357 | 0.585 |
| TN | −0.325 | 0.558 | 100.000 | 0.577 | 0.488 | 0.735 | 0.121 | −0.179 | 0.828 |
| CEC | −0.437 | 0.868 | 0.577 | 100.000 | 0.392 | 0.473 | 0.285 | −0.364 | 0.647 |
| P | −0.445 | 0.490 | 0.488 | 0.392 | 100.000 | 0.436 | −0.002 | −0.051 | 0.433 |
| $C_{stock}$ | −0.168 | 0.392 | 0.735 | 0.473 | 0.436 | 100.000 | −0.135 | 0.246 | 0.820 |
| GWC | −0.076 | 0.309 | 0.121 | 0.285 | −0.002 | −0.135 | 100.000 | −0.693 | 0.259 |
| BD | 0.158 | −0.357 | −0.179 | −0.364 | −0.051 | 0.246 | −0.693 | 100.000 | −0.292 |
| SSI | −0.264 | 0.585 | 0.828 | 0.647 | 0.433 | 0.820 | 0.259 | −0.292 | 100.000 |

| 20–40 cm variable | H+Al | TN | SB | CEC | $C_{stock}$ | GWC | SSI | Granul |
|---|---|---|---|---|---|---|---|---|
| H+Al | 100.000 | 0.588 | −0.565 | 0.873 | 0.532 | 0.240 | 0.627 | −0.222 |
| TN | 0.588 | 100.000 | −0.153 | 0.619 | 0.800 | 0.193 | 0.847 | −0.221 |
| SB | −0.565 | −0.153 | 100.000 | −0.092 | −0.001 | 0.076 | −0.112 | −0.015 |
| CEC | 0.873 | 0.619 | −0.092 | 100.000 | 0.641 | 0.335 | 0.690 | −0.277 |
| $C_{stock}$ | 0.532 | 0.800 | −0.001 | 0.641 | 100.000 | 0.349 | 0.877 | −0.410 |
| GWC | 0.240 | 0.193 | 0.076 | 0.335 | 0.349 | 100.000 | 0.128 | −0.794 |
| SSI | 0.627 | 0.847 | −0.112 | 0.690 | 0.877 | 0.128 | 100.000 | −0.061 |
| Granul | −0.222 | −0.221 | −0.015 | −0.277 | −0.410 | −0.794 | −0.061 | 100.000 |

**Table A7.** Correlation coefficients of all soil attributes based on Pearson correlation analysis for each of the depth layers.

| 0–10 cm variable | pH | H+Al | TN | SB | CEC | P | $C_{stock}$ | GWC | BD |
|---|---|---|---|---|---|---|---|---|---|
| pH | 100.000 | −0.880 | −0.359 | 0.933 | −0.398 | −0.356 | −0.163 | −0.227 | 0.056 |
| H+Al | −0.880 | 100.000 | 0.437 | −0.855 | 0.725 | 0.350 | 0.186 | 0.147 | −0.192 |
| TN | −0.359 | 0.437 | 100.000 | −0.215 | 0.528 | 0.592 | 0.735 | 0.227 | −0.079 |
| SB | 0.933 | −0.855 | −0.215 | 100.000 | −0.262 | −0.248 | 0.032 | −0.148 | 0.161 |
| CEC | −0.398 | 0.725 | 0.528 | −0.262 | 100.000 | 0.322 | 0.389 | 0.076 | −0.142 |
| P | −0.356 | 0.350 | 0.592 | −0.248 | 0.322 | 100.000 | 0.631 | −0.071 | 0.205 |
| $C_{stock}$ | −0.163 | 0.186 | 0.735 | 0.032 | 0.389 | 0.631 | 100.000 | −0.071 | 0.462 |
| GWC | −0.227 | 0.147 | 0.227 | −0.148 | 0.076 | −0.071 | −0.071 | 100.000 | −0.363 |
| BD | 0.056 | −0.192 | −0.079 | 0.161 | −0.142 | 0.205 | 0.462 | −0.363 | 100.000 |
| $qCO_2$ | 0.022 | 0.068 | −0.106 | −0.020 | 0.101 | −0.199 | −0.297 | 0.356 | −0.507 |
| qMic | −0.157 | 0.039 | 0.042 | −0.060 | −0.008 | 0.198 | 0.320 | −0.193 | 0.604 |
| BetaGlu | −0.326 | 0.400 | 0.354 | −0.322 | 0.317 | 0.070 | 0.055 | 0.367 | −0.486 |
| Ure | −0.506 | 0.458 | 0.250 | −0.479 | 0.216 | 0.241 | 0.108 | 0.127 | −0.050 |
| FDA | −0.560 | 0.600 | 0.443 | −0.552 | 0.382 | 0.266 | 0.142 | 0.273 | −0.287 |
| EwR | −0.105 | −0.005 | 0.148 | −0.052 | −0.079 | 0.135 | 0.326 | −0.021 | 0.312 |
| EwD | −0.023 | −0.095 | 0.051 | 0.047 | −0.115 | 0.087 | 0.303 | −0.097 | 0.405 |
| EwB | −0.195 | 0.218 | 0.097 | −0.196 | 0.146 | 0.011 | 0.023 | 0.227 | −0.143 |
| $Lit_{Prd}$ | −0.152 | 0.256 | 0.085 | −0.220 | 0.184 | 0.016 | −0.231 | −0.057 | −0.391 |
| $Lit_{Nut}$ | −0.023 | 0.106 | −0.041 | −0.093 | 0.072 | −0.053 | −0.229 | −0.136 | −0.272 |
| SSI | −0.293 | 0.423 | 0.879 | −0.164 | 0.571 | 0.555 | 0.744 | 0.130 | −0.190 |
| Granul | −0.158 | 0.233 | 0.083 | −0.212 | 0.152 | −0.057 | −0.119 | −0.067 | −0.180 |

| variable | $qCO_2$ | qMic | BetaGlu | Ure | FDA | EwR | EwD | EwB | $Lit_{Prd}$ |
|---|---|---|---|---|---|---|---|---|---|
| pH | 0.022 | −0.157 | −0.326 | −0.506 | −0.560 | −0.105 | −0.023 | −0.195 | −0.152 |
| H+Al | 0.068 | 0.039 | 0.400 | 0.458 | 0.600 | −0.005 | −0.095 | 0.218 | 0.256 |
| TN | −0.106 | 0.042 | 0.354 | 0.250 | 0.443 | 0.148 | 0.051 | 0.097 | 0.085 |
| SB | −0.020 | −0.060 | −0.322 | −0.479 | −0.552 | −0.052 | 0.047 | −0.196 | −0.220 |
| CEC | 0.101 | −0.008 | 0.317 | 0.216 | 0.382 | −0.079 | −0.115 | 0.146 | 0.184 |
| P | −0.199 | 0.198 | 0.070 | 0.241 | 0.266 | 0.135 | 0.087 | 0.011 | 0.016 |
| $C_{stock}$ | −0.297 | 0.320 | 0.055 | 0.108 | 0.142 | 0.326 | 0.303 | 0.023 | −0.231 |
| GWC | 0.356 | −0.193 | 0.367 | 0.127 | 0.273 | −0.021 | −0.097 | 0.227 | −0.057 |
| BD | −0.507 | 0.604 | −0.486 | −0.050 | −0.287 | 0.312 | 0.405 | −0.143 | −0.391 |
| $qCO_2$ | 100.000 | −0.608 | 0.266 | −0.086 | 0.126 | −0.207 | −0.237 | 0.069 | 0.125 |
| qMic | −0.608 | 100.000 | −0.272 | 0.068 | −0.039 | 0.304 | 0.358 | −0.055 | −0.354 |
| BetaGlu | 0.266 | −0.272 | 100.000 | 0.332 | 0.571 | −0.077 | −0.147 | 0.139 | 0.094 |
| Ure | −0.086 | 0.068 | 0.332 | 100.000 | 0.605 | 0.012 | −0.034 | 0.148 | 0.006 |
| FDA | 0.126 | −0.039 | 0.571 | 0.605 | 100.000 | 0.083 | −0.019 | 0.336 | 0.128 |
| EwR | −0.207 | 0.304 | −0.077 | 0.012 | 0.083 | 100.000 | 0.884 | 0.493 | −0.254 |
| EwD | −0.237 | 0.358 | −0.147 | −0.034 | −0.019 | 0.884 | 100.000 | 0.277 | −0.302 |
| EwB | 0.069 | −0.055 | 0.139 | 0.148 | 0.336 | 0.493 | 0.277 | 100.000 | −0.025 |
| $Lit_{Prd}$ | 0.125 | −0.354 | 0.094 | 0.006 | 0.128 | −0.254 | −0.302 | −0.025 | 100.000 |
| $Lit_{Nut}$ | 0.061 | −0.355 | −0.060 | −0.118 | −0.051 | −0.226 | −0.258 | −0.068 | 0.923 |
| SSI | 0.010 | −0.085 | 0.377 | 0.191 | 0.413 | 0.141 | 0.048 | 0.119 | 0.142 |
| Granul | 0.086 | −0.156 | −0.043 | 0.071 | 0.193 | −0.073 | −0.090 | −0.056 | 0.524 |

| variable | $Lit_{Nut}$ | SSI | Granul |
|---|---|---|---|
| pH | −0.023 | −0.293 | −0.158 |
| H+Al | 0.106 | 0.423 | 0.233 |
| TN | −0.041 | 0.879 | 0.083 |
| SB | −0.093 | −0.164 | −0.212 |
| CEC | 0.072 | 0.571 | 0.152 |
| P | −0.053 | 0.555 | −0.057 |

**Table A7.** *Cont.*

| variable | Lit$_{Nut}$ | SSI | Granul |
|---|---|---|---|
| C$_{stock}$ | −0.229 | 0.744 | −0.119 |
| GWC | −0.136 | 0.130 | −0.067 |
| BD | −0.272 | −0.190 | −0.180 |
| qCO$_2$ | 0.061 | 0.010 | 0.086 |
| qMic | −0.355 | −0.085 | −0.156 |
| BetaGlu | −0.060 | 0.377 | −0.043 |
| Ure | −0.118 | 0.191 | 0.071 |
| FDA | −0.051 | 0.413 | 0.193 |
| EwR | −0.226 | 0.141 | −0.073 |
| EwD | −0.258 | 0.048 | −0.090 |
| EwB | −0.068 | 0.119 | −0.056 |
| Lit$_{Prd}$ | 0.923 | 0.142 | 0.524 |
| Lit$_{Nut}$ | 100.000 | 0.054 | 0.501 |
| SSI | 0.054 | 100.000 | 0.238 |
| Granul | 0.501 | 0.238 | 100.000 |

**10–20 cm**

| variable | pH | H+Al | TN | SB | CEC | P | C$_{stock}$ | GWC | BD | SSI | Granul |
|---|---|---|---|---|---|---|---|---|---|---|---|
| pH | 100.000 | −0.721 | −0.325 | 0.745 | −0.437 | −0.445 | −0.168 | −0.076 | 0.158 | −0.264 | 0.073 |
| H+Al | −0.721 | 100.000 | 0.558 | −0.616 | 0.868 | 0.490 | 0.392 | 0.309 | −0.357 | 0.585 | −0.138 |
| TN | −0.325 | 0.558 | 100.000 | −0.197 | 0.577 | 0.488 | 0.735 | 0.121 | −0.179 | 0.828 | −0.160 |
| SB | 0.745 | −0.616 | −0.197 | 100.000 | −0.145 | −0.355 | −0.032 | −0.162 | 0.134 | −0.141 | −0.061 |
| CEC | −0.437 | 0.868 | 0.577 | −0.145 | 100.000 | 0.392 | 0.473 | 0.285 | −0.364 | 0.647 | −0.212 |
| P | −0.445 | 0.490 | 0.488 | −0.355 | 0.392 | 100.000 | 0.436 | −0.002 | −0.051 | 0.433 | −0.194 |
| C$_{stock}$ | −0.168 | 0.392 | 0.735 | −0.032 | 0.473 | 0.436 | 100.000 | −0.135 | 0.246 | 0.820 | −0.327 |
| GWC | −0.076 | 0.309 | 0.121 | −0.162 | 0.285 | −0.002 | −0.135 | 100.000 | −0.693 | 0.259 | −0.018 |
| BD | 0.158 | −0.357 | −0.179 | 0.134 | −0.364 | −0.051 | 0.246 | −0.693 | 100.000 | −0.292 | −0.075 |
| SSI | −0.264 | 0.585 | 0.828 | −0.141 | 0.647 | 0.433 | 0.820 | 0.259 | −0.292 | 100.000 | −0.052 |
| Granul | 0.073 | −0.138 | −0.160 | −0.061 | −0.212 | −0.194 | −0.327 | −0.018 | −0.075 | −0.052 | 100.000 |

**20–40 cm**

| variable | pH | H+Al | TN | SB | CEC | P | C$_{stock}$ | GWC | BD | SSI | Granul |
|---|---|---|---|---|---|---|---|---|---|---|---|
| pH | 100.000 | −0.697 | −0.347 | 0.817 | −0.359 | −0.191 | −0.214 | −0.014 | 0.403 | −0.324 | 0.070 |
| H+Al | −0.697 | 100.000 | 0.588 | −0.565 | 0.873 | 0.208 | 0.532 | 0.240 | −0.414 | 0.627 | −0.222 |
| TN | −0.347 | 0.588 | 100.000 | −0.153 | 0.619 | 0.286 | 0.800 | 0.193 | −0.305 | 0.847 | −0.221 |
| SB | 0.817 | −0.565 | −0.153 | 100.000 | −0.092 | −0.106 | −0.001 | 0.076 | 0.323 | −0.112 | −0.015 |
| CEC | −0.359 | 0.873 | 0.619 | −0.092 | 100.000 | 0.188 | 0.641 | 0.335 | −0.308 | 0.690 | −0.277 |
| P | −0.191 | 0.208 | 0.286 | −0.106 | 0.188 | 100.000 | 0.257 | −0.055 | −0.103 | 0.285 | 0.012 |
| C$_{stock}$ | −0.214 | 0.532 | 0.800 | −0.001 | 0.641 | 0.257 | 100.000 | 0.349 | −0.013 | 0.877 | −0.410 |
| GWC | −0.014 | 0.240 | 0.193 | 0.076 | 0.335 | −0.055 | 0.349 | 100.000 | 0.008 | 0.128 | −0.794 |
| BD | 0.403 | −0.414 | −0.305 | 0.323 | −0.308 | −0.103 | −0.013 | 0.008 | 100.000 | −0.401 | −0.176 |
| SSI | −0.324 | 0.627 | 0.847 | −0.112 | 0.690 | 0.285 | 0.877 | 0.128 | −0.401 | 100.000 | −0.061 |
| Granul | 0.070 | −0.222 | −0.221 | −0.015 | −0.277 | 0.012 | −0.410 | −0.794 | −0.176 | −0.061 | 100.000 |

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
