# Peer review of "Traditional Yerba Mate Agroforestry Systems in Araucaria Forest in Southern Brazil Improve the Provisioning of Soil Ecosystem Services"

_conservation, doi:10.3390/conservation4010009_

Round 1

Reviewer 1 Report

Comments and Suggestions for Authors

The authors of the article entitled "" demonstrated the greater potential of traditional yerba mate agroforestry systems in providing ecosystem services, via higher quality soil. The data is important to demonstrate that production systems can be aligned with maintaining the provision of ecosystem services. In general, the results need to be better discussed and the differences between management systems need to be tested with more rigorous procedures, which go beyond descriptive statistics.

The more detailed comments are in the attached file.

Author Response

Dear Reviewer
We thank you for your work and for suggesting improvements on the original version of the manuscript entitled ‘Traditional yerba mate agroforestry systems in Araucaria Forest in Southern Brazil improve soil ecosystem services provisioning’. 
We have made changes to it, following your comments. We consider that all the changes were addressed with the revised version and hope that you will agree with the improvements.
Best regards
The authors

Reviewer 2 Report

Comments and Suggestions for Authors

All in the MS

Comments on the Quality of English Language

All in the MS

Author Response

(The authors gave the same response as above.)

Reviewer 3 Report

Comments and Suggestions for Authors

Congratulations on the work presented.

The manuscript is very well-structured and easily understood. The introduction is comprehensive, and for someone unfamiliar with this agroecosystem, it's easy to grasp its structure and importance.

However, you might consider improving the format of pages where only one figure appears, such as pages 10 and 12 of the document, for instance. Perhaps you could reduce the size of the figures to fit multiple ones on the same page along with their captions. Lastly, under the caption of Figure 5, there is a paragraph with a font size that appears to be smaller than it should be; please review it.

In conclusion, except for the aforementioned formatting corrections, it seems that this document requires no further changes.

Author Response

Dear Reviewer

We are sending a letter explaining that there was no self-plagiarism but rather greater exploration of the data.

Best regards

Authors

Round 2

Reviewer 1 Report

Comments and Suggestions for Authors

The manuscript is suitable for publication. I only suggest a detailed review for the correct spelling of the measurement units.

Author Response

Dear Editors of Conservation Journal

Dear reviewer

We have made a detailed revision of the spelling of measurement units and all errors found were corrected. Thanks for the care.

We also made a small change at the end of the article title, which does not change the meaning of the sentence, but makes it clearer.

Best regards

The authors
